# High-throughput screening of BAM inhibitors in native membrane environment

Parthasarathi Rath[1], Adrian Hermann [2], Ramona Schaefer[2], Elia Agustoni [1], Jean-Marie Vonach[2], Martin Siegrist[2], Christian Miscenic[2], Andreas Tschumi[2], Doris Roth[2], Christoph Bieniossek[2] ✉ & Sebastian Hiller [1] ✉

The outer membrane insertase of Gram-negative bacteria, BAM, is a key target for urgently needed novel antibiotics. Functional reconstitutions of BAM have so far been limited to synthetic membranes and with low throughput capacity for inhibitor screening. Here, we describe a BAM functional assay in native membrane environment capable of high-throughput screening. This is achieved by employing outer membrane vesicles (OMVs) to present BAM directly in native membranes. Refolding of the model substrate OmpT by BAM was possible from the chaperones SurA and Skp, with the required SurA concentration three times higher than Skp. In the OMVs, the antibiotic darobactin had a tenfold higher potency than in synthetic membranes, highlighting the need for native conditions in antibiotics development. The assay is successfully miniaturized for 1536-well plates and upscaled using large scale fermentation, resulting in high-throughput capacities to screen large commercial compound libraries. Our OMV-based assay thus lays the basis for discovery, hit validation and lead expansion of antibiotics targeting BAM.

Antibiotics against novel targets are urgently needed in this era of rising multidrug resistance[1,2]. A particularly promising target is the β-barrel assembly machinery (BAM) complex, which is located at the cellular periphery, directly accessible from the extracellular space[3,4]. The *E. coli* BAM complex consists of the integral β-barrel protein BamA and the four associated lipoproteins BamB, BamC, BamD and BamE[5,6]. BAM folds and inserts outer membrane proteins (OMPs) as substrates into the membrane. In their folded forms, these substrates mediate numerous physiological functions including nutrient transport, secretion, cell adhesion and signaling[7,8], making the OMP biogenesis pathway essential and thus a suitable target for novel antibiotics[9,10]. Besides the membrane insertase BAM, the biogenesis pathway involves multiple additional chaperones that guide the OMPs from their point of synthesis in the cytoplasm across the periplasm towards the outer membrane[11,12]. In particular, the periplasmic chaperones Skp and SurA prevent OMP aggregation in the periplasm by keeping the OMPs in highly dynamic conformational ensembles[13–15].

Previous studies have introduced protocols to reconstitute BAM-mediated OMP folding in vitro[6,16,17]. Individual BAM components or the entire complex were expressed, purified and reconstituted into synthetic membrane environments such as proteoliposomes or lipid bilayer nanodiscs to then study the insertion of suitable substrates into the membrane. For example, a recent study investigated BAM-dependent folding of EspP, OmpA and OmpG in vitro with such a reconstitution[16]. Other studies used the integral β-barrel membrane protease OmpT as folding substrate[18–20]. The aspartyl protease activity of folded and membrane-inserted OmpT then allows cleavage of a self-quenched reporter peptide, generating a fluorescence readout to monitor insertion yields. Such assay has been described in proteoliposomes and lipid bilayer nanodiscs[6,17,21].

Notably, however, the reported reconstitutions employ highly non-physiological buffer conditions of pH 9.5 and 1.75 M urea, or above, making their adaptation towards high-throughput screening questionable. In addition, the use of artificial bilayers represents only a modest approximation to the biophysical properties of the native

[1]Biozentrum, University of Basel, Spitalstrasse 41, 4056 Basel, Switzerland. [2]Roche Pharma Research and Early Development, Roche Innovation Center Basel, F. Hoffmann-La Roche Ltd, Grenzacherstrasse 124, 4070 Basel, Switzerland. ✉e-mail: christoph.bieniossek@roche.com; sebastian.hiller@unibas.ch

Gram-negative bacterial outer membrane, which is an asymmetric assembly with distinct and specific lipid composition in the two leaflets[22]. The outer membrane is comprised of predominantly phospholipids in the inner leaflet and lipopolysaccharide (LPS) in the outer leaflet[23]. All in all, the available functional reconstitutions of BAM-mediated folding have so far not reflected physiological conditions nor accomplished the requirements for a high-throughput format.

Outer membrane vesicles (OMVs) allow the reconstitution of OMPs in physiological environment and have consequently been used for many such proteins with diverse biological functions[24,25]. OMVs emerge spontaneously from Gram-negative bacteria by budding of the outer membrane and subsequent release of spherical vesicles. These vesicles feature the exact outer membrane lipid composition including the LPS in the outer leaflet (Fig. 1a)[26,27]. OMVs play essential roles in bacterial survival during stress response, protein secretion, nutrient acquisition, biofilm development and also promote bacterial communication and pathogenesis[28–32]. In biotechnological applications, OMVs are a proven platform for vaccine development against specific and relevant pathogenic microorganisms[33–37]. Expressing the BAM complex in OMVs overcomes the above limitations of purifying and reconstituting samples.

Here, we sought to develop a functional assay for quantitative studies of BAM-mediated OMP folding under physiological conditions in native membrane that is capable of high-throughput screening for BAM inhibitors. We thus employ OMVs that contain functional BAM complex to establish an assay that monitors OmpT refolding by fluorescence. We optimize the production of BAM-OMVs and the assay reaction parameters. We validate the assay with the natural antibiotic darobactin[9,38] and then demonstrate its upscaling and miniaturization to allow high throughput screening of large compound libraries.

## Results

### OmpT-OMVs mediate cleavage of the QF peptide

As a starting point for the assay development, we characterized the activity of folded OmpT in the native membrane environment. To this end, we produced OMVs enriched with the OmpT protease (OmpT-OMVs) by overexpressing OmpT in *E. coli* BL21 (DE3) Omp8. The Omp8 bacterial strain is devoid of the four major OMPs OmpA, OmpF, OmpC and LamB, and thus prone to hypervesiculation upon expression of outer membrane or periplasmic proteins (Fig. 1a), resulting in OMVs with minimal protein background[39]. Proper folding of OmpT in the OMVs was confirmed by the characteristic heat-shift on SDS-PAGE and expression levels were quantified using SDS-PAGE densitometry (Supplementary Fig. 1a, b). We then set up enzymatic reactions by mixing 0.8 nM OmpT in OmpT-OMVs with variable concentrations of the quenched fluorescent (QF) peptide Abz-Ala-Arg-Arg-Ala-Tyr(NO2)-NH2 and measured the emitted fluorescence over time using a plate reader (Supplementary Fig. 1c). Progress curves were obtained by converting the fluorescence signal into the concentration of cleaved peptide using a hyperbolic calibration equation (Fig. 1b, Supplementary Fig. 1d). From the progress curves, the initial reaction rates as a function of substrate concentration were determined and fitted to the classical Michaelis-Menten equation[40] (Fig. 1c). As expected, OmpT peptidase activity followed Michaelis-Menten kinetics very well, with parameters $K_M = 69 \pm 7\,\mu M$ and $v_{max} = 0.066 \pm 0.003\,\mu M \cdot s^{-1}$. Lineweaver-Burk analysis of the same data yielded $K_M = 97 \pm 2\,\mu M$ and $v_{max} = 0.08 \pm 0.003\,\mu M \cdot s^{-1}$ (Fig. 1d). These values correspond to

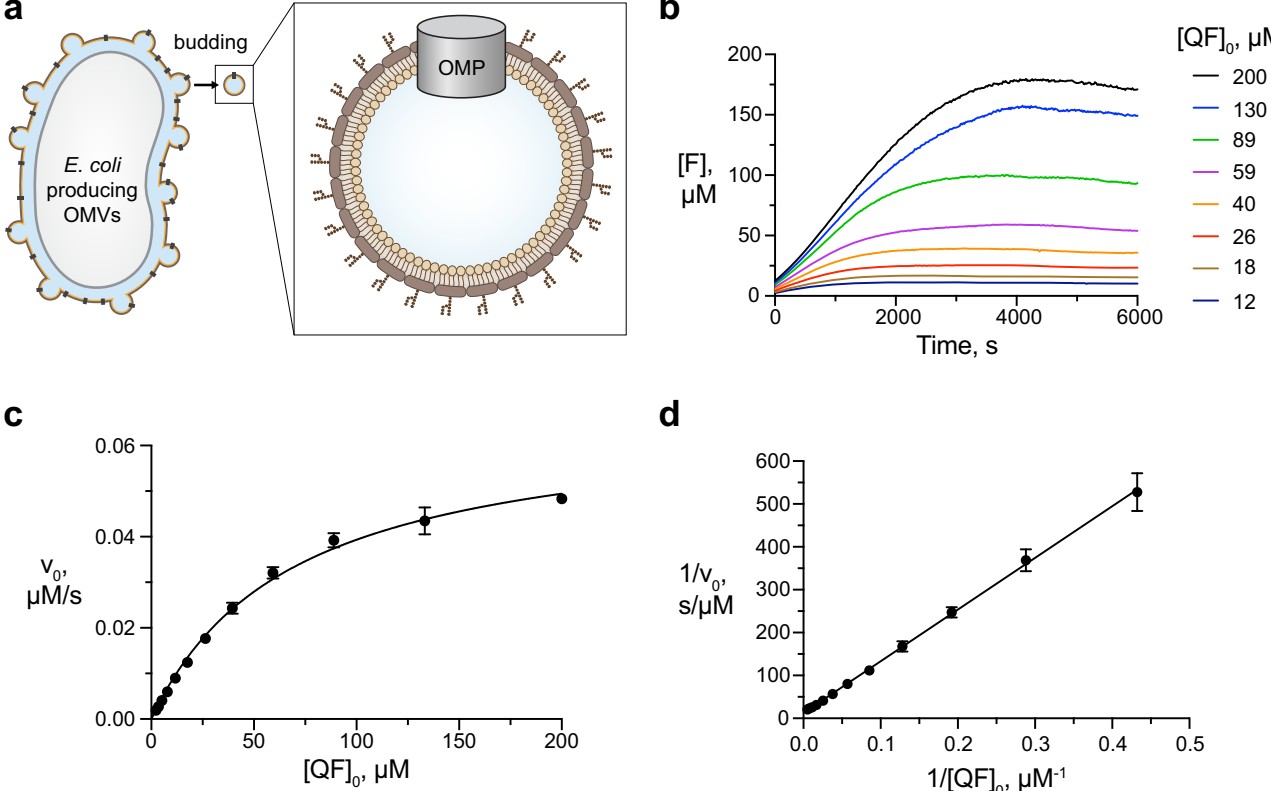

**Fig. 1 | OmpT activity in OMVs. a** Schematic representation of an *E. coli* BL21(*DE3*) Omp8 cell producing OMVs carrying an outer membrane protein (OMP). **b** Progress curves of the OmpT cleavage reaction at variable initial concentrations of QF peptide. [F] = concentration of fluorophore generated by cleavage of the QF peptide. Data set is representative of *n* = 3 independent experiments with similar results. **c** Initial reaction velocities (data points) obtained from (**b**), fitted with the Michaelis-Menten equation (solid line). *n* = 3 independent experiments. Data are presented as average values ± standard deviation. **d** Lineweaver-Burk analysis of the same data. The data underlying panels (**b**–**d**) are provided as Source Data.

turnover numbers ($k_{cat}$) for peptidase activity of OmpT of 93–100 s$^{-1}$ and the OmpT activity is thus around 2–3 times larger than the previously reported value of $k_{cat} = 38$ s$^{-1}$ in micellar environment. This difference may well reflect an optimal functionality in the native-like OMV environment.[20] As a control experiment for subsequent assay development, we confirmed that the presence of the molecular chaperone SurA did not significantly affect the peptidase activity of OmpT, as expected from the respective biological functions (Supplementary Fig. 1e, f). Overall, these results indicate that OmpT is functional in OMVs and that its peptidase activity can be used as a readout for the BAM insertase reaction.

## BAM-mediated OmpT folding in OMVs

To study BAM's insertase activity, we produced OMVs enriched with the BAM complex (BAM-OMVs). This was achieved by simultaneous overexpression of all five BAM complex members from a polycistronic plasmid into the *E. coli* BL21 (DE3) Omp8 strain. In the resulting OMVs, the BAM complex was highly pure and enriched, with typical concentrations of 0.3 mg/mL BAM complex in OMVs (Supplementary Fig. 2a). The BAM-OMVs had a homogeneous size distribution with an average of around 100 nm as determined by cryo-electron microscopy and dynamic light scattering (DLS) (Fig. 2a and Supplementary Fig. 2b). The BamA band on SDS-PAGE exhibited a characteristic gel-shift upon boiling, indicating the proper folding of this transmembrane part of the complex in OMVs (Fig. 2b). Thereby, the presence of LPS in the gel electrophoresis did not have a significant effect on the migration behavior of BamA or the BAM lipoproteins BamB–BamE, as observed

by a comparison with purified BAM complex in DDM micelles, and in full agreement with published experiments on other outer membrane proteins[41]. The integrity of the BAM complex in OMVs was further verified by SEC analysis. BAM-OMVs or BAM-OMVs mixed with SurA-OmpT were solubilized with the detergent DDM and directly applied to size exclusion chromatography. The BAM complex eluted as a unique peak and SDS-PAGE of the elution fractions showed the presence of all five BAM proteins. (Supplementary Fig. 2c–f). The excess amounts of BamC, BamD, and SurA-OmpT eluted separately from the intact BAM complex elution peak (Supplementary Fig. 2e, f). From the band intensities, we estimate that at least 70% of the BamA was integrated in complete BAM complexes in these preparations.

We then aimed to monitor the BAM activity in these BAM-OMVs by fluorescence by a set of two coupled reactions. In the first reaction, unfolded OmpT constitutes the substrate of BAM. Correctly inserted OmpT resulting from the first reaction can then catalyze the second reaction, namely cleavage of the QF peptide, resulting in increased fluorescence emission (Fig. 2c). For the first reaction to occur, the model substrate OmpT needs to reach the entry point for its membrane insertion, the lateral gate of BAM. Since the OMVs have the same topology as the native membrane, this gate is directed towards the OMV lumen and unfolded OmpT coming from the OMV outside thus needs to cross the OMV membrane. One possibility to achieve is the use of a membrane disrupting reagent, such as the membrane-perforating drug colistin (polymyxin E). To test for the impact, OMVs incubated with colistin up to a concentration of 100 μM were characterized by negative-stain electron microscopy and dynamic light

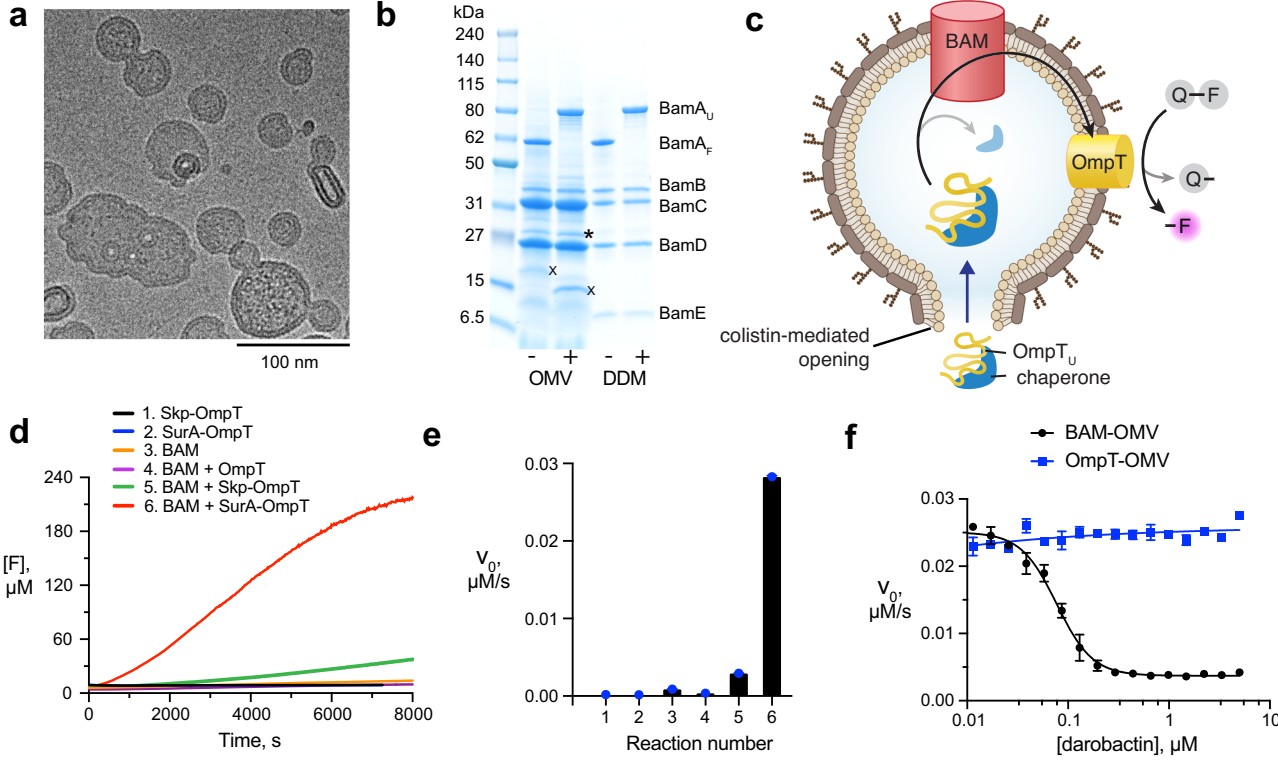

**Fig. 2 | Assay for BAM function in native environment. a** Cryo-electron micrograph of 50 μM colistin-treated BAM-OMVs. The experiment was performed one time. The selected image is representative for the entire EM grid. **b** SDS-PAGE of BAM-OMVs and purified BAM complex in DDM micelles unboiled (−) and boiled (+). The position of folded (BamA$_F$) and unfolded (BamA$_U$) is indicated, along with the lipoproteins BamB–BamE. The bands labelled with asterisk and X, correspond to truncated BamC and OmpX, respectively, as identified by mass spectrometry. The uncropped gel is provided as Source Data. The experiment was repeated independently 3 times with similar results. **c** Schematic representation of the assay setup comprising BAM-OMVs, colistin, chaperone-bound OmpT and QF peptide. **d** Progress curves for six different reaction mixes 1–6 as indicated. Data set is representative of $n = 3$ independent experiments with similar results. **e** Initial reaction velocities from D. $n = 3$ independent experiments. Data are presented as average values ± standard deviation. **f** Inhibitory effect of darobactin on BAM-mediated OmpT folding in OMVs. Data points indicate initial reaction velocities. The fitted curve corresponds to an $IC_{50}$ value of 85 ± 7 nM. $n = 2$ independent experiments. Data are presented as average values ± standard deviation. The data underlying panels (**d**–**f**) are provided as Source Data.

scattering (DLS). No significant perturbation in average size distribution of the OMVs was detected and BamA remained well folded at all tested colistin concentrations as evidenced by SDS-PAGE gel shift (Supplementary Fig. 3a–g). The use of colistin for the assay was thus deemed feasible. Finally, the unfolded OmpT must be transported in suitable form across the aqueous solution to its insertion point, which can be achieved by molecular chaperone, such as periplasmic Skp or SurA. Overall, the functional assay thus comprises BAM-OMVs pre-incubated with 20 μM colistin, OmpT–chaperone complexes and the QF peptide. The combination of these components results in a coupled reaction scheme that translates BAM insertase function into a fluorescence readout (Fig. 2c).

The BAM functional assay in OMVs was validated with specific control reactions (Fig. 2d–e, Supplementary Fig. 4). First, we verified that unfolded OmpT in presence of either of the chaperones but in the absence of OMVs ('Skp-OmpT' and 'SurA-OmpT') was inactive and thus did not cleave the QF peptide. This is in full agreement with the expectation that the membrane protein OmpT cannot fold in the absence of a membrane mimetic. Second, a reaction mixture comprising BAM-OMVs only, but in absence of unfolded OmpT ('BAM') resulted in negligible QF peptide cleavage, which corresponds to the presence of minimal endogenous levels of OmpT in the OMVs. Third, a reaction of BAM-OMVs with unfolded OmpT, but in the absence of the chaperones Skp and SurA ('BAM-OmpT') resulted in no QF peptide cleavage compared to the BAM-OMVs only, which indicated that OmpT insertion is not catalyzed by BAM in absence of chaperones. Presumably, the insoluble OmpT forms amorphous aggregates under these conditions. Fourth, OMVs not enriched in BAM (empty OMVs or mCherry-OMVs) were similarly produced as of BAM-OMVs. Empty OMVs were obtained by harvesting the endogenous OMVs from Omp8 cells without overexpression of any protein. mCherry OMVs were obtained by overexpressing mCherry into the periplasm of the Omp8 cells. Both these OMVs do not contain detectable amounts of the BAM complex (Supplementary Fig. 4a). Equimolar amounts of empty OMV and mCherry OMV showed negligible activity in presence of SurA-OmpT (Supplementary Fig. 4b). Finally, in contrast to these four negative control reactions, substantial QF peptide cleavage was observed in the complete assay setup, i.e., when chaperone-bound OmpT was provided to BAM-OMVs ('BAM-Skp-OmpT' and 'BAM-SurA-OmpT'). Notably, in this assay the chaperone SurA had a higher catalyzing efficiency than Skp.

Additional controls were done by replacing wild type OmpT with the catalytically impaired mutant OmpT(G216K,K217G) or the catalytically defect mutant OmpT(D105A). Consequently, the assay showed reduced and no activity, respectively (Supplementary Fig. 4b). Applying the reaction mixture in unboiled form in SDS-PAGE showed a gel shift of folded OmpT compared to unfolded OmpT, further confirming the folding of OmpT by BAM in the reaction (Supplementary Fig. 4c). Finally, we tested the role of each of the four BAM lipoproteins on BAM-mediated OmpT folding. A knockout plasmid was generated for each lipoprotein and the resulting BAM lipoprotein-deficient complex was produced in OMVs and quantified in the assay setup (Supplementary Fig. 5a). Each BAM lipoprotein-deficient complex showed a significantly reduced activity compared to the complete BAM complex (Supplementary Fig. 5b, c). These data clearly indicate that OmpT is delivered to the BAM complex in a chaperone-dependent manner, properly folded, and inserted into the outer bacterial membrane. Furthermore, none of the four lipoproteins BamB–BamE are essential, but each contributes quantitatively to the refolding efficiency.

We concluded assay validation by studying the inhibitory effect of the natural antibiotic darobactin. Darobactin inhibits BAM via binding to the lateral gate of BamA, thus preventing access of substrates[9]. We pre-incubated colistin-treated BAM-OMVs with variable concentrations of darobactin for 10 min at 37 °C. Subsequently, the QF peptides and SurA-OmpT mixture were added to the reaction mix and the fluorescence emission was monitored. The inhibitory effect of darobactin is clearly visible and a control experiment showed that OmpT activity was not affected by darobactin, i.e. that the compound in fact hit the first of the coupled reactions (Fig. 2f and Supplementary Fig. 6a–c). Remarkably, in the outer membrane environment, darobactin had an inhibitory concentration ($IC_{50}$) of $85 \pm 7$ nM, which is more than one order of magnitude lower than in an artificial membrane environment[9]. This finding clearly demonstrates the validity of the assay for high-throughput screening and at the same time highlights the importance of the native membrane for antibiotic development. The possibility to counter-screen with OmpT-OMVs provides a method to distinguish BAM from OmpT inhibitors.

## Optimization of reaction parameters

To determine the optimal colistin concentration, a fixed concentration of BAM-OMVs with and without sonication was incubated with variable colistin concentrations for 10 min and then subjected to the reaction mixture with QF peptide and SurA–OmpT or Skp–OmpT (Fig. 3a, Supplementary Fig. 7a–c). The largest assay activity was observed in the range of 20–50 μM colistin. Concentrations larger than 100 μM led to a significant activity reduction, presumably by too severe perturbation of the membranes. Notably, activity of the assay was non-zero in the absence of colistin, pointing to the presence of imperfections in the OMVs, such as holes, ruptures, and inversions that allow access to BAM. At the intermediate colistin concentrations, the OMVs are still largely intact, as evidenced by cryo- and negative stained electron microscopy and dynamic light scattering (DLS) (Supplementary Fig. 3). Finally, we tested whether sonication could also be used instead of colistin. Indeed, sonication of BAM-OMVs in absence of colistin resulted in increased activity, showing independently that access to BAM is limited by membrane topology in the untreated OMVs. Since sonication is not well amenable in large volumes, colistin treatment presented the method of choice for the subsequent high-throughput setup.

Next, the optimal pH range was tested, both for the coupled BAM–OmpT reaction setup as well as for OmpT alone (Fig. 3b, Supplementary Fig. 7d–f). The activity of OmpT alone in OmpT-OMVs as a function of pH shows a maximum at pH 6.5, in excellent agreement with the previously reported pH-dependence of OmpT activity in detergent micelles[20]. The optimal activity of the coupled assay is pH 5.5 for Skp-bound and pH 8.0 for SurA-bound OmpT. We therefore continued all experiments systematically at a pH 7.2.

As chaperones play a critical role in the assay, we investigated the concentration dependences of Skp and SurA. We incubated a fixed concentration of 2 μM OmpT with variable Skp and SurA concentrations for 30 min at room temperature prior to the experiment and then applied them to the reaction mixture containing a fixed concentration of colistin-treated BAM-OMVs and QF peptide. The reaction rate increased with increasing Skp or SurA concentrations, reaching the maximum at 50 μM for Skp and 200 μM for SurA (Fig. 3c, Supplementary Fig. 8a, b). This result showed that in our experimental conditions, a stoichiometric ratio of 25:1 for Skp:OmpT and 100:1 for SurA:OmpT, or above is required for maximal folding efficiency of OmpT by BAM-OMVs. Furthermore, the data indicate that the insertion rate at early time points is nearly four times higher in the case of SurA-mediated OmpT folding compared to Skp-mediated folding at saturating concentrations. This is in good agreement with the known higher affinity of OMPs for Skp[42], which corresponds to a lower release rate. We were also interested to study competitive effects between the two periplasmic chaperones. The addition of excess amounts of SurA to a reaction mixture containing 50 μM Skp did not change OmpT's folding activity (Fig. 3d, Supplementary Fig. 8c). On the other hand, addition of Skp to a reaction mixture containing 15 μM SurA resulted in an increase of the initial rate of the reaction with simultaneous decrease of the overall activity of OmpT (Fig. 3e, Supplementary Fig. 8d). These

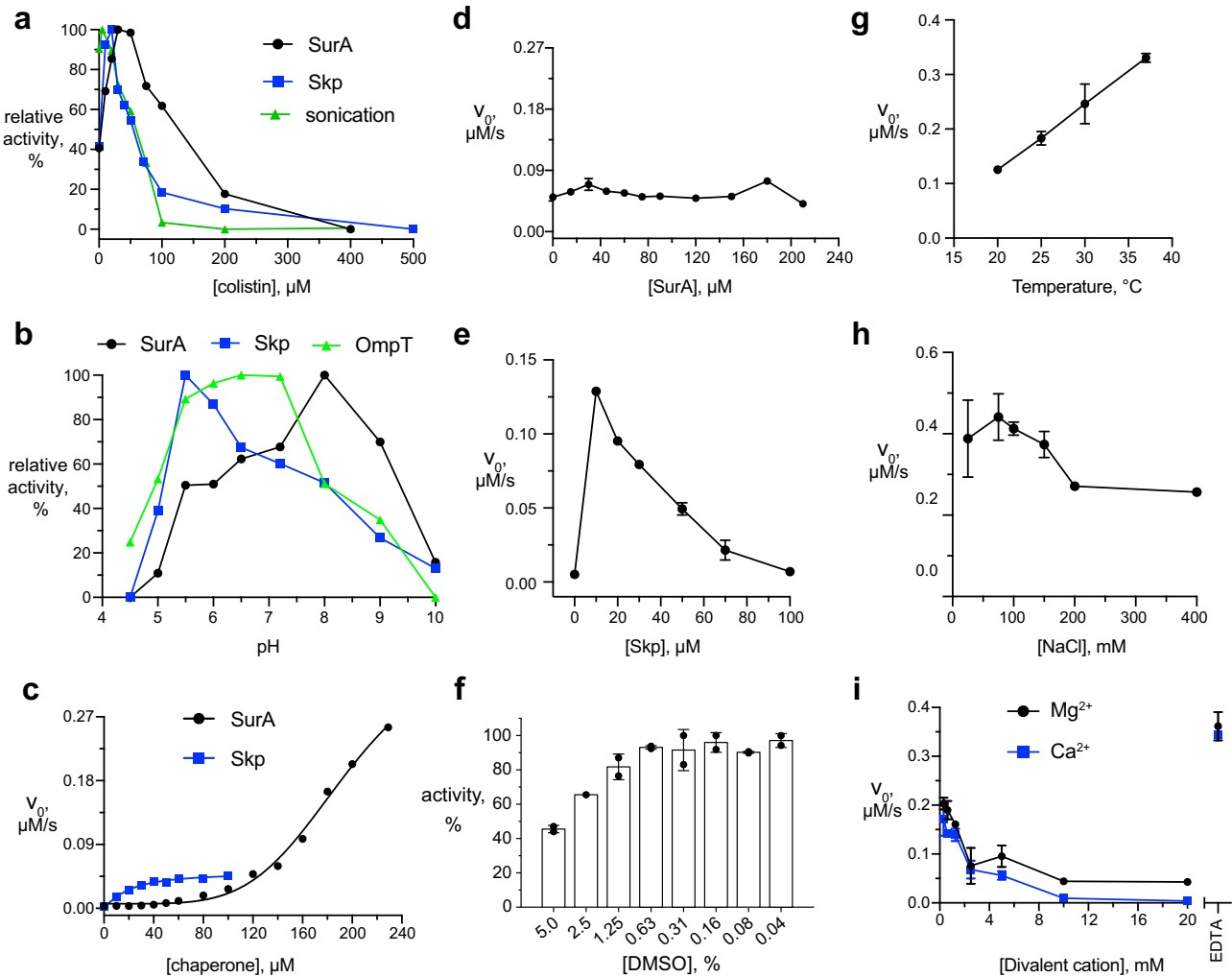

**Fig. 3 | Optimization of reaction parameters. a** Relative activities of BAM-mediated OmpT folding in presence of Skp (blue) and SurA (black) as a function of colistin concentration. **b** Relative activities of pre-folded OmpT (green) as well as BAM-mediated OmpT folding in presence of Skp (blue), SurA (black), or for sonicated OMVs (green), as a function of pH value. **c** Initial reaction velocities of BAM-mediated OmpT folding as a function of SurA (black dots) and Skp (blue squares) concentrations. **d** Initial reaction velocities of BAM-mediated OmpT folding at 50 μM Skp plus variable concentrations of SurA. **e** Initial reaction velocities of BAM-mediated OmpT folding at 15 μM SurA plus variable concentrations of Skp. **f** Normalized activity at variable DMSO concentration. Initial reaction velocities of BAM-mediated OmpT folding at variable (**g**) reaction temperature, (**h**) NaCl, and (**i**) divalent cation concentrations. For each of panels (**d**–**i**), $n = 2$ independent experiments and data are presented as average values ± standard deviation. The measurements underlying panels (**a**–**i**) are shown in Supplementary Figs. 7–9. The data underlying panels (**a**–**i**) are provided as Source Data.

findings are in full agreement with the higher affinity of Skp for OmpT than SurA, resulting in a dominance of Skp over SurA in binding the substrate OmpT in aqueous solution.

Next, the effect of the reaction temperature, and the concentrations of salt and divalent cations on the activity of BAM-mediated OmpT folding were evaluated, because these can affect outer membrane integrity[43,44]. Increase of the reaction temperature resulted in a linear increase in initial reaction velocity in the range 20–37 °C (Fig. 3g, Supplementary Fig. 9a). This is the expected Arrhenius effect on reaction kinetics. The reaction was not detectably affected by increasing NaCl concentrations up to 150 mM (Fig. 3h, Supplementary Fig. 9b), suggesting that electrostatic interaction do not play a dominant role in the rate-limiting steps of the coupled reaction. Finally, a negative effect of the divalent cations (Mg²⁺ and Ca²⁺) was observed. At concentrations of 1.25 mM and above, the assay activity was drastically reduced and the activity was completely inhibited at concentrations of 10 mM and above. The inverse effect, an increase in activity was observed upon complete neutralization of such ions from the reaction mixture by addition of EDTA (Fig. 3i, Supplementary Fig. 9c, d). Since

divalent ions are known to stabilize the LPS bilayer, these data suggest that high divalent ion concentration rigidify the membrane to prevent efficient insertion of the substrate.

Last but not least, as compound libraries are commonly solubilized in DMSO, the robustness of the assay against DMSO was analyzed. The assay was completely unaffected by concentrations up to 0.6% DMSO and only mildly affected up to 5% DMSO. Only for DMSO concentrations above 5% the activity dropped markedly (Fig. 3f). Typical compound library screening setups operate at final DMSO concentrations of 1%, making the assay sufficiently robust.

## High-throughput screening of BAM inhibitors

Target-based screening in the search for antibacterial compounds remains challenging, as the translatability towards cell-based activity is often limited. The OMV assay presenting BAM in the outer bacterial membrane combines the possibility of screening directly against an essential bacterial target with a setting close to the native Gram-negative bacterial cell with minimal background of other proteins. Having established optimal reaction conditions, we therefore aimed at

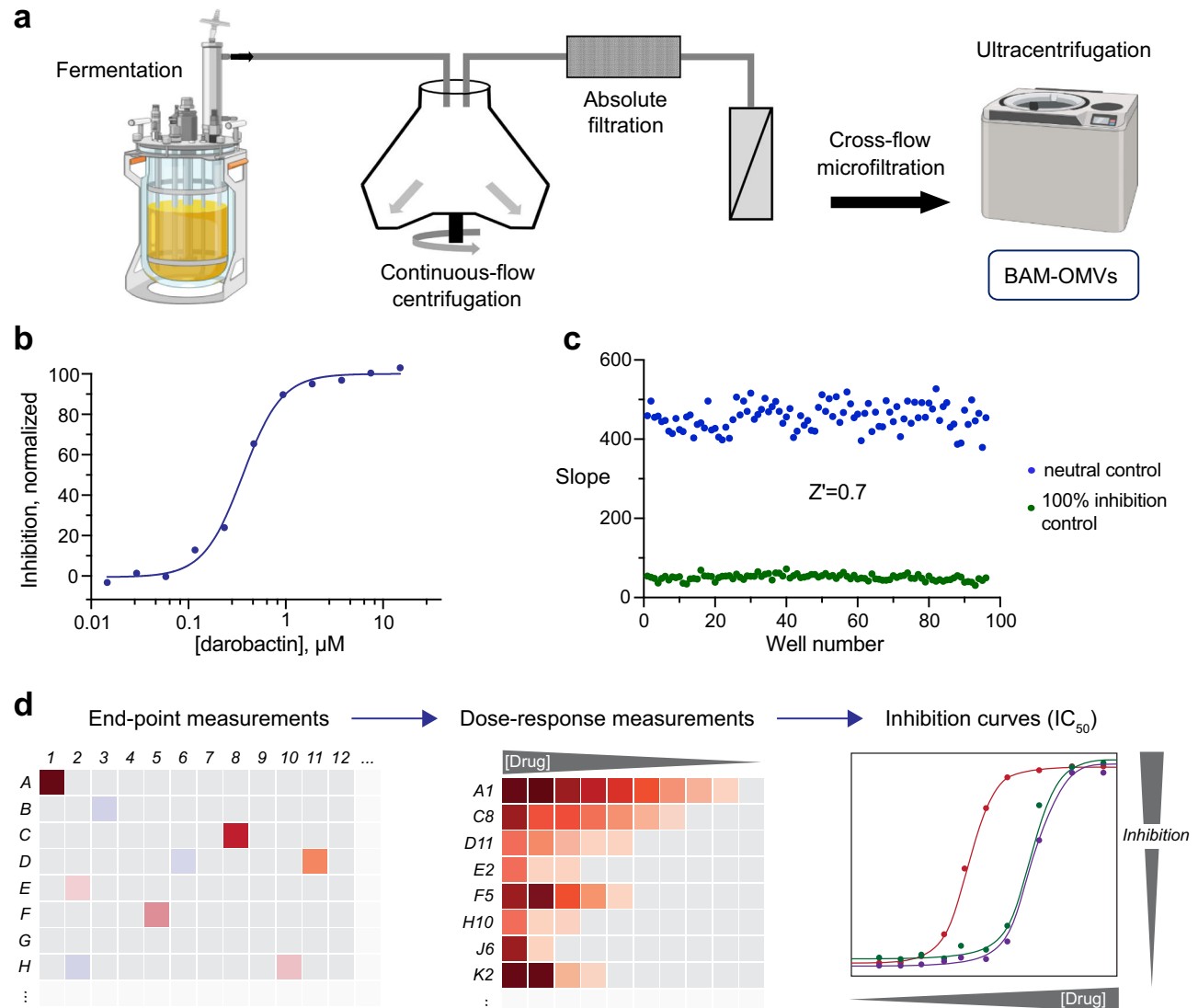

**Fig. 4 | Platform for high-throughput screening of BAM inhibitors. a** Scheme for upscaling of BAM-OMV production. BAM-OMV expression is carried out in a 100 L fermentation culture in LB medium. After expression, LB cultures are passed through a continuous-flow centrifugation step for harvesting. The supernatant is then filtered to eliminate remaining cell debris and intact cells, and concentrated 25-fold by cross-flow microfiltration with a 30 kDa cut-off. The concentrate is then ultracentrifuged to collect BAM-OMVs which are further washed, flash-frozen in liquid nitrogen and stored at −80 °C. **b** Dose response curve of BAM-mediated

OmpT folding in BAM-OMVs at variable darobactin concentration. The $IC_{50}$ of darobactin is 0.23 μM. **c** Slopes of neutral (blue) and 100% inhibition control (green) reactions carried out in an Aurora 1536 well plate, representing a robust sensitivity window with a z' factor of 0.7. **d** Schematic work flow for high-throughput screening for BAM inhibitors. An end-point measurement is done for all compounds, then a dose-response curve is recorded for initial hits and inhibition constants are calculated. The data underlying panels (**b**–**c**) are provided as Source Data. Figure 4a was created with BioRender.com.

developing the assay towards the capacity to screen inhibitors in high-throughput format.

To achieve a cost-effective capacity of the order of tens of thousands and potentially up to 1 million compounds including downstream dose-response measurements, both the miniaturization of the individual reactions, along with upscaling the production of all components, most notably the OMVs, is required. The upscaling of BAM-OMV production was achieved by using a 100 L fermenter. For adapting the purification protocol accordingly, we identified two major challenges. First, separation of the OMVs from intact bacterial cells in such large volumes and, second, high-speed ultra-centrifugation of the 100 L supernatant where volumes of maximal 400 mL can be processed in typical centrifuges.

To address these points, in a first step bacterial cells were separated from the OMVs by continuous-flow centrifugation. Thereby, the isolation process was optimized to minimize contamination of the

OMV fraction by bacterial cells. The resulting supernatant was then further cleared from remaining bacterial cells and cell debris via a microfiltration step. The cleared flow-through was concentrated to approximately 4 L by home-build cross-flow microfiltration similar to the procedure previously described by the Kuehn lab[45] (Fig. 4a). This highly BAM-OMV enriched supernatant was suitable for ultra-centrifugation and stored in 200–400 mL fractions at −80 °C for subsequent processing. BAM-OMVs generated by this process were found to be fully functional and of the same quality as material from small scale isolations. Inhibition by darobactin was tested and displayed a mean $IC_{50}$ of 230 nM with a standard deviation of 48 nM in the miniaturized assay format (Fig. 4b).

Miniaturization of assay volumes, on the other hand, was required to (i) reduce the overall consumption of components and to (ii) increase the overall throughput by switching to a 1536-well setting in combination with the use of a robotic system. As today's

state-of-the-art acoustic dispensing systems allow spotting compounds in low nL volumes, we aimed for overall reaction volumes of as little as 6 µL/well. Assay optimization resulted in volumes of 4 µL of a mixture of BAM-OMVs and colistin solution. This mixture was added to each well and the plates were then incubated for 15 min at 37 °C. Subsequently, 2 µL of a mixture of OmpT, SurA and QF peptide was added to start the BAM-dependent folding reaction. The plate was gently centrifuged and the emission fluorescence at 430 nm (excitation wavelength 360 nm) was immediately measured for reference in each well using a microplate reader. After 20 min incubation at 37 °C, emission fluorescence in each well was recorded again (end-point measurement). The reaction rate was calculated by subtracting the start value from the end-point value. In this setting the assay displayed a robust window with a z′ factor of 0.7 (Fig. 4c). Thus, the assay fulfilled all specifications for high-throughput screening and was utilized in this setting to screen approximately 10,000 compounds, although screening in the million-compound range is also possible without changing settings. Within the 10,000 compounds we identified 39 molecules displaying at least 50% of inhibition. 23 resynthesized hits were subsequently re-analyzed in a dose response setting using the same assay format with varying compound concentrations, keeping the overall DMSO concentration constant at 1% (Fig. 4d). Best hits displayed up to 90% of inhibition with an $IC_{50}$ of about 40 µM while not showing direct OmpT inhibition or membrane disrupting activity. Notably, the test library was limited by its overall purity and overall chemical diversity and therefore does not reflect a differentiated chemical space and potential hit rates.

## Discussion

In this study, we have employed OMVs as a platform to reconstitute BAM-mediated OMP folding in its native environment in vitro. We have optimized biochemical parameters of the system and scaled the assay up for high-throughput screening. Finally, the assay proved versatile to screen large inhibitor libraries. Notably, while the complex assay due to the coupled reaction nature can deliver inhibitors for both BAM and OmpT, the latter can be readily counter-screened using OmpT-OMVs. For minimizing interference in the screening process, the assay could be further optimized by changing the fluorogenic peptide to a peptide labeled with a higher wavelength fluorophore and a corresponding quencher.

In addition to high-throughput screening, the functional assay in OMVs will be very useful to address specific biological questions of OMP biogenesis in the future. We have given two examples in the present work, where we quantified the contribution of individual BAM subunits and studied the competitive relationship of the two chaperones Skp and SurA and observed the dominance of Skp. The latter behavior is readily explained by the higher affinity of the OmpT substrate to the trimeric Skp cavity compared to the more open binding site on SurA[14,42,46–48]. Furthermore, the assay can also be extended to study OMVs produced by other Gram-negative strains or E. coli variants, for example to probe the effect on LPS variations on BAM function, or to understand the functional role of other bacterial BamA homologues.

The importance of native environment for antibiotic development is highlighted by the present findings on the natural antibiotic darobactin. Darobactin binds to the lateral gate to inhibit insertion and further folding of OmpT[9,38]. In reconstituted proteoliposomes, BamA inhibited folding of OmpT with an $IC_{50}$ and the minimal inhibitory concentration (MIC) value of around 1 µM under in vitro growth conditions[9]. Here, we have observed that the direct inhibitory concentration in native environment can be as low as 80 nM, i.e., the compound binds in these conditions with an affinity that is at least one order of magnitude stronger. This can be rationalized by BamA being slightly perturbed in the artificial membrane and populating the energetically most favorable conformation only in the full native

membrane. Importantly, this finding suggests that the MIC value of darobactin in vivo is most likely not limited by the affinity to its binding site but rather by secondary effects such as solubility or permeability. The latter can likely be improved by modifications of the compound, suggesting the potential for further improvement of this compound class.

Antimicrobial resistance by the ESKAPE group of pathogens is a major threat to human health worldwide[49]. Novel antibiotics targeting the outer membrane insertase of Gram-negative bacteria with the major component BamA are particularly promising to counteract in this situation[50,51]. With the high throughput inhibitor screening platform developed in this study, screening of large compound libraries has become possible, which is likely to help the identification and optimization of classes of antibiotics targeting Gram-negative pathogens via inhibition of the BAM complex.

## Methods

### Expression and preparation of BAM- and OmpT-OMVs

OMVs containing either the BAM complex or OmpT were produced as previously reported[39]. In brief, expression was carried out in the E. coli BL21(DE3)Omp8 strain[52]. A polycistronic plasmid carrying all BAM complex subunits and ampicillin resistance was used for BAM production in OMVs. Freshly transformed cells were grown in 2–4 L cultures of LB medium containing 100 µg/mL ampicillin at 37 °C. Protein expression was induced at $OD_{600nm}$ of 0.4 with 1 mM IPTG. Expression was ended after 2 h at an $OD_{600nm}$ of 0.9–1.0, before the stationary phase was reached, because OMVs collected from late stationary phase contain high amounts of impurities[39,53]. At this time point, cell suspensions were centrifuged at 10,000 g for 15 min. The supernatant was collected, filtered through a 0.22 µM diameter membrane filter (Millipore) and then ultracentrifuged at 200,000 g for 1 h, with the OMVs ending up in the pellet. The pellet was resuspended in sodium phosphate buffer, pH 7.2, pelleted again and then resuspended in 400 µL of the same buffer, flash-frozen in liquid nitrogen and stored at −80 °C. Endogenous OMVs (empty-OMVs) were harvested without stimulation of vesiculation. OMVs carrying overproducing the red fluorescent protein mCherry were obtained by overexpressing mCherry with a periplasmic targeting sequence. Plasmids for overexpression of BAM complex variants deficient of an individual lipoprotein were generated starting from the polycistronic plasmid by introducing a stop codon at the beginning of each gene of interest. BAM-OMVs deficient of each lipoprotein were then produced as described above.

### Expression and purification of Skp and SurA

Expression and purification of Skp and SurA were carried out as previously reported[13,15]. Briefly, E. coli BL21 (DE3) Lemo cells were transformed with a pET28b-derived vector carrying either Skp or SurA with N-terminal (His)6-tag and TEV cleavage. Cells were grown in LB medium containing 50 µg/mL kanamycin and 30 µg/mL chloramphenicol at 37 °C. At $OD_{600nm}$ of 0.7, growth temperature was reduced to 25 °C, and cells were induced with 0.4 mM IPTG for overnight expression. Cells were then harvested by centrifugation and resuspended in lysis buffer containing 25 mM HEPES, pH 7.5, 300 mM NaCl and lysozyme, DNAse and PMSF. Cells were lysed in a high-pressure homogenizer. The lysate was centrifuged at 24,000 g, 4 °C for 30 min and the soluble fraction was filtered using a 0.45 µM diameter membrane filter (Sarstedt). The lysate was loaded on a His-trap HP column (Cytiva), washed, and then eluted with 500 mM imidazole. The fractions containing Skp or SurA were dialyzed overnight against 4 L of 25 mM HEPES, pH 7.5, 150 mM NaCl at 4 °C. Proteins were then unfolded with 6 M guanidine hydrochloride (Gn-HCl), and loaded on high affinity Ni-NTA resin for 2 h at 4 °C. Skp or SurA were eluted in an unfolded form in 6 M Gn-HCl and refolded by dialysis at 4 °C. 1 mM DTT and 0.5 mM EDTA were added to the protein solution for overnight TEV cleavage reaction at 4 °C. Another dialysis step was performed against dialysis buffer for

2 h. The protein solutions were again unfolded with 6 M Gn-HCl, and incubated with high affinity Ni-NTA resin for 2 h at 4 °C. Flow-through and washings containing (His)$_6$-tag-free Skp or SurA were collected and again refolded by dialysis against dialysis buffer for overnight at 4 °C. Refolded proteins were concentrated by ultrafiltration, and loaded on a HiLoad 16/600 Superdex 200 pg column (Cytiva) pre-equilibrated with 25 mM MES buffer, pH 6.5 and 100 mM NaCl. Elution fractions corresponding to pure Skp or SurA were concentrated, quantified, flash-frozen and stored at −80 °C.

### Expression and purification of unfolded BamA and OmpT

BamA, OmpT, or the OmpT mutants G216K,K217G and D105A were expressed in inclusion bodies following the protocol above, except that the cell pellets obtained from harvesting the cultures were used. Cells homogenized in lysis buffer containing 20 mM Tris-Cl, pH 8.0, DNase, PMSF and lysozyme were incubated at room temperature for 20 min and then lysed in a high-pressure microfluidizer. Lysed cells were centrifuged at 12,000 $g$ for 10 min to collect the inclusion bodies, which were washed in 20 mM Tris-Cl, pH 8.0 with 0.1% Triton-X 100 and then dissolved in 6 M Gn-HCl. The solution was centrifuged at 12,000 $g$ for 10 min at room temperature and the supernatant was dialyzed overnight in a 10 kDa membrane filter against water. The membrane protein of interest precipitated out during this dialysis and was then dissolved in 20 mM Tris-Cl, pH 8.0 containing 7 M urea. Unfolded BamA and OmpT were further purified by cation exchange chromatography using a HiTrap Q-HP column (Cytiva) and eluted in a gradient of 1 M NaCl in 20 mM Tris-Cl, pH 8.0 containing 7 M urea. Purified proteins were directly flash-frozen in liquid nitrogen and stored at −80 °C.

### SDS-PAGE densitometry, dynamic light scattering (DLS) and electron microscopy

The amounts of BamA and OmpT in OMVs were quantified by comparison with samples of known concentrations of urea-unfolded BamA or OmpT. Each sample was boiled for 10 min at 98 °C prior to loading on 4–20% SDS-PAGE (BioRad). SDS-PAGE gels were stained in Coomassie blue and then destained for analysis. The intensities of the protein bands were quantified in ImageJ. To quantify the amount of BAM complex, BAM-OMVs were solubilized with 1% DDM and applied on a 4 mL Superdex 200 gel filtration column (Cytiva) equilibrated with 20 mM Tris-Cl, pH 8.0, 150 mM NaCl containing 0.05% DDM. To obtain the size distribution of OMVs by DLS measurements, OMV preparations were diluted 100-fold in sodium phosphate buffer, pH 7.2. The mean light scattering was calculated from 10 independent measurements, each recorded at 25 °C with a 10 s acquisition time. For negative-stained EM, 10-fold diluted OMV samples were stained with 2% uranyl acetate and images were collected at 80 kV with a nominal magnification of 105 k. For Cryo-EM, samples were flash frozen in liquid ethane and micrographs were acquired at 200 kV in a FEI Talos F200C TEM microscope.

### OmpT activity assay

OmpT activity was monitored by proteolytic cleavage of the quenched fluorescent (QF) peptide Abz-Ala-Arg-Arg-Ala-Tyr(NO$_2$)-NH$_2$ (GenScript). This peptide is self-quenched and therefore non-fluorescent in its pure form, but becomes fluorescent upon cleavage by OmpT. Reaction mixtures consisting of variable concentrations of OMVs, chaperones, OmpT, colistin and QF peptide were prepared in 20 mM Tris-Cl buffer pH 6.5 in Greiner 384-well plates. Reaction mixtures were mixed in orbital mode for 60 s using the shaker installed in a Tecan Spark multi-mode microplate reader. Fluorescence emission at 410 nm was measured at 37 °C with excitation at 320 nm at regular intervals of 20 s. Reactions were carried out in triplicates for analysis.

To calibrate for the non-linearities in the fluorescence detection, a series of reactions with concentrations of QF peptides from 2.3 to 200 µM were mixed with 0.8 nM OmpT and incubated for 2 h to reach complete digestion of the QF peptide. The resulting detected relative fluorescence units (RFU) depend in non-linear fashion on the peptide concentrations used and the data were fitted by the hyperbolic equation

$$E = \frac{A \cdot [QF]}{B + [QF]} \qquad (1)$$

where E is the fluorescence emission, [QF] is the concentration of QF peptide, and A and B are empirical constants. The time-dependence of [QF] was used to determine initial reaction velocities from 0 to 500 s of each reaction, which were analyzed via the classical Michaelis-Menten equation, resulting in the parameters $K_M$ and $k_{cat}$[40]. The effect of SurA on OmpT peptidase activity was carried out in reaction mixes comprised of 8.4 nM OmpT, 100 µM QF peptide in presence of variable concentrations of SurA from 8.3 to 120 µM.

### BAM-mediated OmpT activity assay and parameter optimization

BAM-mediated OmpT activity was assessed in 50 µL reaction mixtures containing 16 nM BAM in OMVs, 20 µM colistin, 500 µM QF peptide, 2 µM unfolded OmpT and 50 µM Skp or SurA. BAM-OMVs were initially mixed with colistin, incubated for 10 min, and aliquoted into 384-well plates. QF peptide solution and pre-incubated Skp–OmpT or SurA–OmpT complexes were added to initiate BAM-mediated OmpT folding and resulting activity. For control reactions, individual components were omitted and replaced by buffer. All reactions were carried out in duplicate. Variation of pH values from 4.5 to 10 was achieved in suitable buffers containing either sodium citrate, sodium phosphate, Tris-HCl or glycine. The final composition of the reactions was 0.8 nM OmpT-OMVs, 50 µM colistin, and 100 µM QF peptide. The catalytic activity of BAM complex in OMVs as a function of pH was measured similarly with a reaction mixture comprising 16 nM BAM in OMVs, 50 µM colistin, 50 µM Skp or SurA, 2 µM unfolded OmpT, and 500 µM QF peptide. Skp and SurA were mixed with unfolded wild type OmpT or one of the OmpT mutants G216K,K217G or D105A and incubated for 10 min prior to be added to the reaction mixture. Activity assays were carried out in equivalent molar concentrations of empty-OMVs, mCherry-OMVs, or OMVs containing BAM deficient of individual lipoproteins BamB–BamE. For the optimization of chaperone concentrations, variable concentrations of Skp (0–80 µM) and SurA (0–226 µM) were mixed with 2 µM of unfolded OmpT and incubated for 30 min at room temperature. The resulting chaperone–OmpT solutions were then mixed with the reaction mixture to a final concentration of 16 nM BAM in OMVs, 100 µM colistin and 500 µM QF peptide.

For the evaluation of the effect of sonication and colistin concentrations, BAM-OMVs were sonicated for 30 min in a bath sonicator operating at 80% frequency at 30 °C. Activity was then measured using an equivalent amount of sonicated and non-sonicated BAM-OMVs at various colistin concentrations (0–400 µM), keeping the remaining reaction parameters constant: 16 nM BAM in OMVs, 50 µM SurA, 2 µM unfolded OmpT, and 500 µM QF peptide. BAM-mediated OmpT folding were further optimized at variable reaction temperatures (37, 30, 25, and 20 °C), salt concentration (25–400 mM NaCl), and divalent cation (20–0.312 mM MgCl$_2$ and CaCl$_2$) concentration.

Size exclusion chromatography followed by SDS-PAGE was used to observe the integrity of BAM complex after completion of the folding reaction. 100 µL BAM-OMVs (4 µM) alone and a reaction mixture comprising of 100 µL BAM-OMVs (4 µM), 20 µM colistin, 250 µL SurA (413 µM) and 10 µL (430 µM) urea unfolded OmpT were incubated at 37 °C for 60 min. 1% DDM was added to the sample and further incubated at 37 °C for 30 min for solubilization. The samples were centrifuged at 20,000 $g$ for 10 min in a bench top centrifuge. The

completely soluble and transparent supernatant was loaded on 4 mL bed volume manually packed Superdex S200 column (for BAM-OMV alone) and on a 120 mL bed volume Superdex 16/60 S200 column (for BAM-OMV reaction mixture) preequilibrated with 20 mM Tris-Cl, pH 8.0, 150 mM NaCl containing 0.05% DDM. Elution fractions were loaded on SDS-PAGE to observe intact BAM complex separately from SurA-OmpT.

## Assay validation

Colistin-mixed BAM-OMVs or sonicated BAM-OMVs in absence of colistin were incubated with variable concentrations of darobactin (11–5000 nM) for 10 min at 37 °C. Darobactin was produced in-house using published protocols[9]. QF peptide and SurA–OmpT were added to initiate the folding reaction. The final reaction mixture contained 10 nM BAM in OMVs, 20 μM colistin, 100 μM QF peptide, 50 μM SurA and 2 μM OmpT. Reactions were carried out in sodium phosphate buffer, pH 7.2. As negative control, OmpT-OMVs were incubated with variable concentrations of darobactin (11–5000 nM) for 10 min at 37 °C. QF peptide was added to the reaction mixture to initiate the folding reaction. The final reaction mixture contained 0.8 nM OmpT, 20 μM colistin and 100 μM QF peptide. Fluorescence emission was monitored at regular intervals of 30 s for approx. 2 h at 37 °C. Initial reaction rates were plotted against the darobactin concentration to obtain $IC_{50}$ values.

## High-throughput screening

The production of BAM-OMVs was scaled up using a fermenter of 100 L expression culture in LB media containing 100 μg/mL ampicillin. A starter culture was grown from a freshly transformed plate, then inoculated to the 100 L fermenter for expression. BAM-OMV production was induced at $OD_{600nm}$ 0.4 with 1 mM IPTG for 2 h at 37 °C. Cells were then passed through a continuous-flow centrifugation at 16,000 g for harvesting. The supernatant was then passed through an absolute filter of 0.22 μM membrane to separate the OMVs from remaining cell debris and intact cells. The filtered supernatant containing BAM-OMVs was concentrated to approximately 4 L through a 30 kDa membrane cut-off using a cross-flow microfiltration step. The concentrated solution was ultracentrifuged at 200,000 g for 1 h to collect the BAM-OMVs which were further washed with Dulbecco's Phosphate Buffered Saline (DPBS buffer), pH 7.2, collected, flash-frozen in liquid nitrogen and stored at −80 °C until further characterization.

To make the screening assay cost-effective, the assay was miniaturized to reaction volumes of 6 μL in Aurora 1536-well plates. The reaction mixture contained 80 nM BamA in OMVs, 20 μM colistin, 50 μM SurA, 2 μM unfolded OmpT and 100 μM QF peptide in the final reaction. The required large quantities of SurA were produced without cleavage of the His-tag, which was shown to have very similar activity compared to His-tag cleaved SurA. Unfolded OmpT in 7 M urea solution was produced as described above. The robustness of the assay against DMSO concentrations in the range 0.039–5% was tested. Fluorescence detection parameters were optimized for maximal sensitivity. Dose-response curves were determined and the data was fitted to a standard Hill curve. Curves were normalized relative to the baselines at high and low inhibitor concentrations.

## Reporting summary

Further information on research design is available in the Nature Portfolio Reporting Summary linked to this article.

## Data availability

The experimental data that support the findings of this study is shown in the article and its supplementary materials. The raw data underlying all Figures and Supplementary Figures is provided as a Source Data file. Source data are provided with this paper.

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

## Acknowledgements

We thank Timothy Sharpe, Mohamed Chami and Carola Alampi for technical support and discussions. This work was supported by the Swiss National Science Foundation via the NRP 72 (Grant 167125 to S.H.) and the National Center of Competence in Research AntiResist (180541). P.R. acknowledges a personal fellowship from the University of Basel Research foundation.

## Author contributions

P.R., A.T., C.B., and S.H. designed the study; P.R. performed expression and production of OMVs and protein reagents, designed and cloned BAM-lipoprotein mutants, assay development, biochemical, biophysical and EM measurements, and supported assay transfer to HTS format; A.H. expressed and purified protein; R.S. and D.R. established the assay transfer to HTS platform and performed the high-throughput screen; C.M., J.M.V. and M.S. established and performed large scale production and purification of OMVs; E.A. modeled kinetics; C.B. and S.H. supervised and coordinated research; all authors analyzed and interpreted data; P.R. and S.H. wrote the manuscript with input from all authors.

## Competing interests

The authors declare no competing interests.
