## [Peer Review File · Nature Communications]

High-throughput screening of BAM inhibitors in native membrane environmentReviewers' Comments:

Reviewer #1:

Remarks to the Author:

The work by Rath et al. involves the development of a BAM overproduction method that will enable high-throughput screening of inhibitors for this essential bacterial complex. While the study is designed well and presented concisely, a few important experimental factors and controls are currently omitted or not elaborated. Without these vital checkpoints in place, the study fails to address the most essential objective, which is to generate a functional BAM complex. Hence, the submission cannot be recommended in its current form.

Major:

1. Is the over-production of BAM under a polycistronic mRNA causing differential expression levels of the various BAM components? This is evident in Figure 2B, wherein the levels of BamC and BamD are highest, and BamB is the lowest. This differential expression will influence the overall amounts of "functional BAM" in the OMVs, thereby negating the whole purpose of this study. Since the efficient functioning of BAM depends on the stoichiometry of its constituents, the authors must show direct read-outs that clearly establish the levels of fully reconstituted BAM in their OMVs.
2. The very important control reaction with OmpT, SurA/Skp, and colistin-permeated OMVs, without the over-produced BAM, is missing. For a study that appears to be through, it is puzzling that the authors overlooked including this very important control.
3. The basal activity of OmpT (Figure 3A, Extended Data Figure 4A-B) is sizeable even in the absence of colistin (at least 40%, as seen in Figure 3A), indicating that OmpT retains the ability to fold in the reaction even without access to the "periplasmic side" of the BAM complex. One could argue that the increase in OmpT activity at low colistin concentrations is due to increase in the levels of folded OmpT, which can be promoted by membrane perturbation. Hence, a major component of the measured fluorescence could indeed be due to the levels of folded OmpT. The experiments must demonstrate 0% OmpT folding in OMVs without BAM, or must be done with a protein other than OmpT.

Minor (page numbers refer to those in the PDF):

1. Is the intrinsic permeability of colistin-treated LUVs sufficiently large to allow a large complex such as Skp/SurA-OmpT to diffuse through? Please comment.
2. Lines 122-128: How does the OmpT activity assayed here compare with other reported values from native or native-like environments?
3. Line 156: please explain "fold of BamA". If the authors are referring to the amount of BamA in the OMV, how was this estimated?
4. Lines 164-180: While this may not be crucial, please include a control demonstrating the effect SurA/Skp may have on cleavage of QF peptide. Similarly, a control to demonstrate that colistin does not influence the binding of darobactin can be considered.
5. Line 185: change "fluorescent emission" to "fluorescence emission".
6. How was darobactin obtained for the study? Please indicate.
7. Line 280: change 10'000 to 10,000.
8. Figure 2B and Extended Data Figure 2I: Were the gels run in the presence of OMVs? If so, the authors must comment on how the presence of LPS did not interfere in the electrophoretic mobility of their samples. The concentrations of the samples appear to be sufficiently high for visualizing with Coomassie blue.
9. Figure 2B and Extended Data Figure 2I: Other outer membrane proteins are generally observed in native preparations, even in the Omp8 variant of E. coli BL21(DE3). However, they are not visible in this preparation.
10. A figure similar to Figure 2B can be included for OmpT obtained directly in OMVs, and after completion of the BAM-mediated folding reaction.
11. Figure 4B: How was the inhibition normalized? Please provide this detail.
12. Extended Data Figure 5: In the figure legend, please indicate that colistin is included in these preparations.

Reviewer #2:

Remarks to the Author:

The outer membrane of Gram-negative bacteria is a unique permeability barrier that prevents toxic molecules such as antibiotics from accessing targets in the cell. Transmembrane beta-barrel outer membrane proteins (OMPs) are folded and inserted into the outer membrane by the beta-barrel assembly machine (BAM) complex. The BAM complex is an attractive antibiotic target due to its essentiality, conservation, and localization in the outer membrane. Multiple inhibitors targeting the BAM complex have been discovered through phenotypic screens and in vitro binding assays. However, the field is lacking a robust activity assay that captures BAM in its native outer membrane context to enable high-throughput screens for inhibitors of OMP folding that can potentially be developed into much-needed antibiotics.

In their manuscript entitled "High-throughput screening of BAM inhibitors in native membrane environment", Hiller and co-workers describe a new, outer membrane vesicle (OMV)-based OMP folding assay, validate this approach with a known BAM inhibitor (darobactin), and scale up this method to enable high-throughput screening. The work described is both timely given the need for approaches to discover new antibacterial molecules and useful for potentially providing insight into the molecular mechanisms of OMP biogenesis, a fundamental process in Gram-negative bacteria. The manuscript is well-written and understandable, experiments and results are described clearly, and the presented data support the conclusions. Though the impact of this work would be increased with additional information about the described high-throughput screen, this manuscript will be of interest to the readers of Nature Communications, especially to microbiologists, investigators searching for antibiotics, and those interested in membrane protein folding.

Here are some specific (some quite minor) questions that arose during the reading of this manuscript:

-Line 128 – Has OmpT activity been measured in a whole cell assay as it would be informative to compare this with the activity observed in the OMV assay? In vitro OmpT activity assays have characterized defective OmpT mutants (e.g., G216K/K217G): could the effect of such mutants in the OMV-based assay also be informative as to the role of a native outer membrane environment on OMP function?

-Line 129 – Suggest wording change from "...that OmpT is well functional in OMVs..." to "...that OmpT is functional in OMVs..."

-Line 137 – Are the authors able to determine the numbers of BAM complexes per OMV? From the practical standpoint, I wonder if the folding is uniform across the OMVs in the assays or if a small number of OMVs with multiple BAM are responsible for the activity? It is more of a basic question, but I wonder 1) if all of the OMVs are sampling similar, uniform portions of the outer membrane environment, and 2) if BAM is forming islands, as observed for other OMPs, and how this affects OMP folding.

-Line 151 – Despite the scalability difficulties, was an OMV sonication approach tested at a small scale to compare with the colistin approach for OmpT uptake? Were similar results observed?

-Line 157 – In Extended Data Figure 2F-I it does appear (to this reader at least) that there is a slight shift away from the smaller diameter OMVs at higher concentration of colistin. It is unlikely to affect the findings, but I wonder if this is significant and if it reveals something about the effect of colistin on the outer membrane.

-Line 159 – "...achieved by a molecular chaperones, such as the periplasmic Skp or SurA." should be

"...achieved by molecular chaperones, such as periplasmic Skp or SurA."

-Line 161 – Were the authors able to assess how efficient uptake of the OmpT-chaperone complex was? Was OmpT-chaperone uptake uniform across the OMV sample and consistent from assay to assay? Did larger OMVs capture more OmpT-chaperone?

-Line 174 – Were the authors able to determine whether, in the absence of chaperones, OmpT was not active because it did not get into the OMVs or because it got into the OMVs but could not be inserted into the outer membrane?

-Line 174 – The controls described are appreciated. Though making OMVs lacking BamA is difficult due to the essentiality of BamA, could the authors make OMVs from cells depleted for BamA to demonstrate reduced activity with such OMVs?

-Line 210 – Is there a known explanation for the different pH optimums for Skp-bound substrate (5.5) versus SurA-bound substrates (8.0)? Does this have biological implications?

-Line 238 – The assay optimizations and characterization are appreciated. A few other conditions that could be biologically revealing are temperature, salt concentration, and divalent cation concentration as these can all affect the outer membrane, and potentially BAM function. Testing the effects of divalent cations might be especially important for screening applications as this could affect both the colistin-mediated uptake step as well as the state of the outer membrane in the OMVs, and could also be altered by certain compounds.

-Line 260 – Maybe your approach is slightly different, but I would suggest citing work by the Kuehn lab here as they have described the use of tangential flow concentrations to facilitate purification of large batches of OMVs previously.

-Line 265 – The Darobactin IC₅₀ with the HTS OMV preparation was 360 nM while that reported with the small-scale preparation (Line 190) was 85 nM. Is this within the range of prep-to-prep variability or does it represent a tangible difference in the preparations?

- Line 267 – "...consumption of components in to (ii) increase..." should probably be "...consumption of components and to (ii) increase..."

-Line 280-284 – The application of this assay to a high-throughput screen is very exciting. This reviewer was left wanting quite a bit more description of the reported experiment with 10,000 compounds. For example:

oWhat was the hit rate of the initial screen?

oWhat was the range of activities?

oHow many compounds from the primary screen repeated in the dose-response?

oDid any hits inhibit OmpT activity (or some other step in the assay)?

oDid any hits destroy/disrupt the OMVs (a potential for non-specific membrane disruptors)?

oWere any BAM-specific hits identified?

oWhat were the hits (even if the authors are unable to reveal the actual chemical structures, more information about the compounds would be useful)?

oDid any hits exhibit antibacterial activity against whole cells?

oDid any active, OMP folding inhibitors have Gram-negative specific whole-cell antibacterial activity?

-Line 300-304 – It would be nice to see some of these applications tested in this manuscript. Unless there are technical limitations, these do not seem far out of scope for the current work.

Reviewer #1 (Remarks to the Author):

The work by Rath et al. involves the development of a BAM overproduction method that will enable high-throughput screening of inhibitors for this essential bacterial complex. While the study is designed well and presented concisely, a few important experimental factors and controls are currently omitted or not elaborated. Without these vital checkpoints in place, the study fails to address the most essential objective, which is to generate a functional BAM complex. Hence, the submission cannot be recommended in its current form.

Major:

1. Is the over-production of BAM under a polycistronic mRNA causing differential expression levels of the various BAM components? This is evident in Figure 2B, wherein the levels of BamC and BamD are highest, and BamB is the lowest. This differential expression will influence the overall amounts of “functional BAM” in the OMVs, thereby negating the whole purpose of this study. Since the efficient functioning of BAM depends on the stoichiometry of its constituents, the authors must show direct read-outs that clearly establish the levels of fully reconstituted BAM in their OMVs.

We address this comment with a set of three experiments:

First, we do a comparison of band intensities on SDS-PAGE with purified BAM complex (Fig. 2B, Extended Data Figure 2C-F). The referee is correct in pointing out that band intensities seem to point to some excess of BamC and BamD. Our analysis of the complex composition by size exclusion chromatography (Extended Data Figure 2C, D) shows, however, that in the BAM-OMVs the BAM complexes are mostly complete and that there is only a minor excess of BamC and BamD. Importantly, the presence of excess BamC and/or BamD is not a problem for the application in the present study, because these molecules do neither perturb the functional BAM complexes, nor do they autonomously process the substrate. BAM function relies in an essential fashion on its core protein BamA, and the presence of excess individual other subunits in the OMVs therefore does not impair the assay.

Moreover, all BAM quantifications in our work were done on the basis of the BamA subunit. Second, we now include new experiments that show that deletion of each of the subunits BamB – BamE leads to a quantitative decrease of the assay activity (Extended Data Figure 5). This shows that all BAM subunits are present in the assay and contribute functionally to the observed reaction, even though a portion of the ensemble of BAM complexes present might not be completely assembled.

Third, we note that the BAM-mediated folding reaction can be inhibited by darobactin (Fig. 2F), a known inhibitor of BAM that binds to BamA. This experiment shows that the overall observed reaction is BamA-dependent.

Together, these experiments show that all five subunits BamA – BamE are present in the reaction and contribute to the overall reaction. A possible excess of individual subunits, or partially reduced levels of some component does not affect the result.

Finally, we would like to point out that in the living cell, a situation of non-perfect stoichiometries among the ensemble of BAM complexes might exist as well. To our knowledge, there is no published data that would establish all BAM complexes in living bacterial cells were completely assembled and no subunits were in excess.

2. The very important control reaction with OmpT, SurA/Skp, and colistin-permeated OMVs, without the over-produced BAM, is missing. For a study that appears to be thorough, it is puzzling that the authors overlooked including this very important control.

Thanks for pointing this out. We had actually measured this control reaction, but had not included it in the manuscript. It is now included in two different versions. One version uses “empty OMVs”, which are endogenously produced from Omp8 cells without overexpression of a protein. The other version are “mCherry-OMVs”, which result from overexpression of mCherry in the bacterial periplasm. Both these control experiments show no detectable activity, confirming that the observed activity indeed depends on overexpressed BAM (Extended Data Figure 4A,B).

3. The basal activity of OmpT (Figure 3A, Extended Data Figure 4A-B) is sizeable even in the absence of colistin (at least 40%, as seen in Figure 3A), indicating that OmpT retains the ability to fold in the reaction even without access to the “periplasmic side” of the BAM complex. One could argue that the increase in OmpT activity at low colistin concentrations is due to increase in the levels of folded OmpT, which can be promoted by membrane perturbation. Hence, a major component of the measured fluorescence could indeed be due to the levels of folded OmpT. The experiments must demonstrate 0% OmpT folding in OMVs without BAM, or must be done with a protein other than OmpT.

Thanks for pointing this out. The residual activity in the absence of colistin comes from OMVs that have non-ideal shape, such as featuring a broken membrane or having undergone inversion. We have added text to point out the presence of such non-ideal OMVs more clearly (Line 237). Furthermore, we demonstrate that by using sonication instead of colistin, essentially the same activity is obtained, showing that the reduced activity in the absence of a membrane-perturbing agent is indeed solely an accessibility problem. Furthermore, there are three control experiments showing that the entire observed fluorescence intensity depends on the proposed mechanistic pathway: No activity in the absence of BAM (empty OMVs or mCherry-OMVs; Extended Data Figure 4A,B); no activity with the catalytically dead OmpT mutant D105A (Extended Data Figure 4B); the activity can be inhibited by darobactin (Fig. 2F, Extended Data Figure 6A,B). These experiments thus establish that the entire activity is BAM-mediated and the alternative scenarios suggested by the referee do not apply.

Minor (page numbers refer to those in the PDF):

1. Is the intrinsic permeability of colistin-treated LUVs sufficiently large to allow a large complex such as Skp/SurA–OmpT to diffuse through? Please comment.

We address this question by a comparison of colistin-treated OMVs with sonicated OMVs. Sonication disrupts the OMVs, leading to essentially the same effect as colistin treatment (Fig. 3A). This comparison thus shows that reduced activity of the assay is indeed an accessibility problem and that the colistin-induced permeation is sufficient.

2. Lines 122-128: How does the OmpT activity assayed here compare with other reported values from native or native-like environments?

The OmpT activity reported here is to our knowledge the first and only quantitative measurement of OmpT in a native-like environment. We observe a 3-fold higher k_{cat} value compared to reported *in vitro* values and this may well be due to the native environment. We are not aware of other reported enzymatic constants of OmpT in native or native-like environments. We have attempted to determine OmpT activity in cells (Review-only-Figure 1) and observed a 20-50 fold lower activity. The lower activity in cells is presumably due to a substantial portion of the OmpT molecules not being active and/or not accessible.

Review-only-Figure 1. Comparison of OmpT activity in living *E. coli* cells (left) and OMVs (right). Concentrations of OmpT were estimated by SDS-PAGE densitometry.

3. Line 156: please explain “fold of BamA”. If the authors are referring to the amount of BamA in the OMV, how was this estimated?

Thanks for pointing this out. What was meant was that BamA was properly folded, as evidenced by SDS-PAGE gel shift. We have reworded for clarity.

4. Lines 164-180: While this may not be crucial, please include a control demonstrating the effect SurA/Skp may have on cleavage of QF peptide. Similarly, a control to demonstrate that colistin does not influence the binding of darobactin can be considered.

We have now determined the effect of SurA on OmpT-mediated cleavage and show this as a control reaction (Extended Data Figure 1E, F). As expected, we observe no effect. We also measured the activity of darobactin on sonicated OMVs (Extended Data Figure 6B). Darobactin does inhibit the reaction under those conditions, however with a different Hill coefficient.

5. Line 185: change “fluorescent emission” to “fluorescence emission”.

Thanks, this has been reworded.

6. How was darobactin obtained for the study? Please indicate.

Thanks for pointing this out. Darobactin was produced in-house following published protocols, this is now explicitly mentioned in the Methods section.

7. Line 280: change 10'000 to 10,000.

Has been retyped.

8. Figure 2B and Extended Data Figure 2I: Were the gels run in the presence of OMVs? If so, the authors must comment on how the presence of LPS did not interfere in the electrophoretic mobility of their samples. The concentrations of the samples appear to be sufficiently high for visualizing with Coomassie blue.

LPS and/or other lipids are visible as a “smear” at low molecular weights on SDS-gels run from OMVs (Fig. 2B, Extended Data Figures 2A, 3G, and 5A). This is commonly observed, for example in Arunmanee, W. et al. Proc. Nat. Acad. Sci USA **113**, E5034-E5043 (2016). It does not influence the migration of proteins that may also be present in the same sample. We added a short comment to the text (line 147).

9. Figure 2B and Extended Data Figure 2I: Other outer membrane proteins are generally observed in native preparations, even in the Omp8 variant of E. coli BL21(DE3). However, they are not visible in this preparation.

The Omp8 variant is generally devoid of four main OMPs, but some Omps are still present and can be seen in Figure 2B. We have added a lane with purified BAM complex next to it for comparison. There are two bands seen in BAM-OMVs that are not seen in purified BAM. We identified them by mass spectrometry. They correspond to OmpX (showing heat-shift) and BamC with truncated N-terminus.

10. A figure similar to Figure 2B can be included for OmpT obtained directly in OMVs, and after completion of the BAM-mediated folding reaction.

We have included such Figures in Extended Data Figure 1A and 4C, respectively.

11. Figure 4B: How was the inhibition normalized? Please provide this detail.

Thanks for pointing this out. The data were normalized from the baselines of the Hill curve fit at low and high inhibitor concentrations. This is now better described in the methods. There was indeed a slight numeric error in the script, leading to values larger 100%. This has been fixed.

12. Extended Data Figure 5: In the figure legend, please indicate that colistin is included in these preparations.

Has been included (now Extended Data Figure 8).

Reviewer #2 (Remarks to the Author):

The outer membrane of Gram-negative bacteria is a unique permeability barrier that prevents toxigenic molecules such as antibiotic from accessing targets in the cell. Transmembrane beta-barrel outer membrane proteins (OMPs) are folded and inserted into the outer membrane by the beta-barrel assembly machine (BAM) complex. The BAM complex is an attractive antibiotic target due to its essentiality, conservation, and localization in the outer membrane. Multiple inhibitors targeting the BAM complex have been discovered through phenotypic screens and in vitro binding assays. However, the field is lacking a robust activity assay that captures BAM in its native outer membrane context to enable high-throughput screens for inhibitors of OMP folding that can potentially be developed into much-needed antibiotics.

In their manuscript entitled “High-throughput screening of BAM inhibitors in native membrane environment”, Hiller and co-workers describe a new, outer membrane vesicle (OMV)-based OMP folding assay, validate this approach with a known BAM inhibitor (darobactin), and scale up this method to enable high-throughput screening. The work described is both timely given the need for approaches to discover new antibacterial molecules and useful for potentially providing insight into the molecular mechanisms of OMP biogenesis, a fundamental process in Gram-negative bacteria. The manuscript is well-written and understandable, experiments and results are described clearly, and the presented data support the conclusions. Though the impact of this work would be increased with additional information about the described high-throughput screen, this manuscript will be of interest to the readers of Nature Communications, especially to microbiologists, investigators searching for antibiotics, and those interested in membrane protein folding.

Here are some specific (some quite minor) questions that arose during the reading of this manuscript:

-Line 128 – Has OmpT activity been measured in a whole cell assay as it would be informative to compare this with the activity observed in the OMV assay? In vitro OmpT activity assays have characterized defective OmpT mutants (e.g., G216K/K217G): could the effect of such mutants in the OMV-based assay also be informative as to the role of a native outer membrane environment on OMP function?

Thanks for these suggestions. We have now repeated the experiment with two mutants of OmpT, the suggested mutant G216K,K217G with impaired catalytic activity, as well as the catalytically dead mutant D105A. As expected, the first mutant shows reduced activity in the assay and the second mutant has no residual activity (Extended Data Figure 4B). As discussed with referee 1, we are not aware of published measurements of OmpT activity in living cells. We have attempted to determine this activity (Review-only-Figure 1) and observed a 20-50 fold lower activity. The lower activity is presumably due to a substantial portion of the OmpT molecules not being active and/or not accessible.

Review-only-Figure 1. Comparison of OmpT activity in living *E. coli* cells (left) and OMVs (right). Concentrations of OmpT were estimated by SDS-PAGE densitometry.

-Line 129 – Suggest wording change from “...that OmpT is well functional in OMVs...” to “...that OmpT is functional in OMVs...”

Thanks, has been reworded.

-Line 137 – Are the authors able to determine the numbers of BAM complexes per OMV? From the practical standpoint, I wonder if the folding is uniform across the OMVs in the assays or if a small number of OMVs with multiple BAM are responsible for the activity? It is more of a basic question, but I wonder 1) if all of the OMVs are sampling similar, uniform portions of the outer membrane environment, and 2) if BAM is forming islands, as observed for other OMPs, and how this affects OMP folding.

This would indeed be interesting, but we are not aware of a possibility to measure the distribution of BAM complexes in the OMV preparation. Since the OMV hypervesiculation is stimulated by BAM overexpression, we assume that most OMVs contain at least several BAM complexes.

-Line 151 – Despite the scalability difficulties, was an OMV sonication approach tested at a small scale to compare with the colistin approach for OmpT uptake? Were similar results observed?

Yes, we did compare the two permeation methods of colistin and sonication and found similar overall activities at maximal permeation. This is now reported in Fig. 3A.

-Line 157 – In Extended Data Figure 2F-I it does appear (to this reader at least) that there is a slight shift away from the smaller diameter OMVs at higher concentration of colistin. It is unlikely to affect the findings, but I wonder if this is significant and if it reveals something about the effect of colistin on the outer membrane.

We agree that such a slight shift might be present in the DLS histograms. We can only speculate to what it may originate from, such as a reorganization of smaller particles in the presence of colistin.

-Line 159 – “...achieved by a molecular chaperones, such as the periplasmic Skp or SurA.” should be “...achieved by molecular chaperones, such as periplasmic Skp or SurA.”

Has been reworded.

-Line 161 – Were the authors able to assess how efficient uptake of the OmpT-chaperone complex was? Was OmpT-chaperone uptake uniform across the OMV sample and consistent from assay to assay? Did larger OMVs capture more OmpT-chaperone?

This is an interesting question, which we can however not address at this point. We are not aware of experiments to quantify the uptake of OmpT-chaperon complexes as a function of the individual OMV size.

-Line 174 – Were the authors able to determine whether, in the absence of chaperones, OmpT was not active because it did not get into the OMVs or because it got into the OMVs but could not be inserted into the outer membrane?

Similar to above, we cannot address this question, because we are not aware of an experiment to distinguish between the two cases. OmpT is not soluble in aqueous solution and therefore presumably aggregates in the absence of a molecular chaperone immediately into large insoluble oligomers. These aggregates may then diffuse into the OMVs and may encounter a BAM complex, but presumably, the BAM complex is not able to refold an OmpT molecule out of such an aggregate.

-Line 174 – The controls described are appreciated. Though making OMVs lacking BamA is difficult due to the essentiality of BamA, could the authors make OMVs from cells depleted for BamA to demonstrate reduced activity with such OMVs?

Thanks for pointing this out. This is a similar question as posed also by referee 1. We have now included such controls by producing two types of OMVs without BAM overexpression, empty OMVs and mCherry-OMVs. Both these OMVs show no activity (Extended Data Figure 4A, B).

-Line 210 – Is there a known explanation for the different pH optimums for Skp-bound substrate (5.5) versus SurA-bound substrates (8.0)? Does this have biological implications?

The likely explanation is the different isoelectric point of the two proteins. Skp has a pI of 9.5 and SurA of 6.1. Since the interaction of OmpT with the chaperone includes also ionic contributions, pH values around the pI may be disfavored for the respective chaperone and pH values away from the pI can be expected to yield a better chaperone efficiency, in agreement with the observations reported.

-Line 238 – The assay optimizations and characterization are appreciated. A few other conditions that could be biologically revealing are temperature, salt concentration, and divalent cation concentration as these can all affect the outer membrane, and potentially BAM function. Testing the effects of divalent cations might be especially important for

screening applications as this could affect both the colistin-mediated uptake step as well as the state of the outer membrane in the OMVs, and could also be altered by certain compounds.

We agree and have now included optimization of the temperature, the salt concentration and divalent cations. We do observe a linear increase in reaction velocity with temperature, which can be rationalized with the generally enhancing Arrhenius effect of temperature on reaction kinetics. We observed no significant effect of the NaCl concentration on activity. We observed an inhibitory effect of divalent ions, suggesting that the divalent ions stabilize the LPS layer. These data are now included in Fig. 3 and Extended Data Figure 9.

-Line 260 – Maybe your approach is slightly different, but I would suggest citing work by the Kuehn lab here as they have described the use of tangential flow concentrations to facilitate purification of large batches of OMVs previously.

Thanks for pointing this out, we fully agree and now cite Chutkan, ... Kuehn, Quantitative and qualitative preparations of bacterial outer membrane vesicles. *Methods Mol. Biol.* **966** (2013) (Reference number 45).

-Line 265 – The Darobactin IC50 with the HTS OMV preparation was 360 nM while that reported with the small-scale preparation (Line 190) was 85 nM. Is this within the range of prep-to-prep variability or does it represent a tangible difference in the preparations?

Thanks for pointing this out. This seeming discrepancy is part of the batch-to-batch variation. The measured values of darobactin efficacy in the HTS setup was on average 230 nM with a standard deviation of 48 nM, i.e. the reported value of 360 nM was one of the largest batch values measured and therefore somewhat misleading. This is now better described in the manuscript.

- Line 267 – “...consumption of components in to (ii) increase...” should probably be “...consumption of components and to (ii) increase...”

Thanks, has been reworded.

-Line 280-284 – The application of this assay to a high-throughput screen is very exciting. This reviewer was left wanting quite a bit more description of the reported experiment with 10,000 compounds. For example:

oWhat was the hit rate of the initial screen?

oWhat was the range of activities?

oHow many compounds from the primary screen repeated in the dose-response?

oDid any hits inhibit OmpT activity (or some other step in the assay)?

oDid any hits destroy/disrupt the OMVs (a potential for non-specific membrane disruptors)?

oWere any BAM-specific hits identified?

oWhat were the hits (even if the authors are unable to reveal the actual chemical structures, more information about the compounds would be useful)?

oDid any hits exhibit antibacterial activity against whole cells?

oDid any active, OMP folding inhibitors have Gram-negative specific whole-cell antibacterial activity?

Within the 10,000 compounds we identified 39 molecules displaying at least 50% of inhibition. 23 resynthesized hits were subsequently re-analyzed in a dose-response setting using the same assay format with varying compound concentrations, keeping the overall DMSO concentration constant at 1%. Best hits displayed up to 90% of inhibition with an IC_{50} of about 40 μ M while not showing OmpT inhibition or membrane disrupting activity. Notably, the test library was limited by its overall purity as well as overall chemical diversity and therefore does not necessarily reflect a differentiated chemical space and potential hit rates. This information is now included in the manuscript.

-Line 300-304 – It would be nice to see some of these applications tested in this manuscript. Unless there are technical limitations, these do not seem far out of scope for the current work.

We agree and do now include the proposed experiment with deletions of the individual BAM lipoproteins BamB–BamE. This experiment nicely shows that each of the proteins is contributing quantitatively to the efficiency of the complex, but that none of them is essential for OmpT refolding in OMVs (Extended Data Figure 5).

Reviewers' Comments:

Reviewer #1:

Remarks to the Author:

I do very much appreciate the changes that the authors have made, and for incorporating several of the suggestions, in the revised manuscript. However, the current version does not still address the two major concerns I had:

1. Perhaps my suggestion was not clear to the authors. The very important control reaction containing all of the following – OmpT, SurA/Skp, and colistin permeated OMVs, but without the over-produced BAM – is missing even in the revised Extended Data Figure 4A,B. This must not only be included, but the findings from this experimental condition must also be discussed in the context of the findings from OmpT + SurA/Skp + colistin permeated OMVs + BAM, by comparing the read-outs of both experiments directly.

2. I disagree with the authors' conclusion regarding the basal activity of OmpT. Can the authors clarify why the relative amounts of BamC and BamD in Extended Data Figure 4C is not in proportion with the BAM-OMVs sample of panel 4A? It also appears that in 4C, BAM-OMV samples containing Skp-OmpT and SurA-OmpT do not display the BAM protein bands in the same position. The basal activity measured as RFU in several reactions in 4B are comparable to the highest amount of darobactin used in Extended Data Figure 6A. This again indicates that there is sufficient amounts of basal OmpT activity irrespective of BAM. Therefore, I reiterate that in addition to checking these data, the authors must provide irrefutable evidence of 0% OmpT folding + activity in OMVs without BAM, or run at least one experiment with a protein other than OmpT.

Without these vital checkpoints in place, the study fails to address the most essential objective, which is to generate a functional BAM complex. Hence, the submission cannot be recommended in its current form.

Reviewer #2:

Remarks to the Author:

In their revision of "High-throughput screening of BAM inhibitors in native membrane environment", Rath et al. have addressed the main concerns raised by the reviewers. Overall, this manuscript established a novel approach to screening for inhibitors of an essential activity in Gram-negative bacteria, optimized and validated this assay, and applied it to a test library to demonstrate its potential application. Especially critical to the revision was the inclusion of several additional control and optimization experiments for the assay (especially the data in Fig. 3 and Extended Data Figs. 4 and 5) and the expanded descriptions of the use of Darobactin and the initial small molecule screen. The additions and modification have greatly improved the manuscript and it is appropriate for the target audience of Nature Communications.

Point-by-point response #2

Reviewer #1 (Remarks to the Author):

I do very much appreciate the changes that the authors have made, and for incorporating several of the suggestions, in the revised manuscript. However, the current version does not still address the two major concerns I had:

1. Perhaps my suggestion was not clear to the authors. The very important control reaction containing all of the following – OmpT, SurA/Skp, and colistin permeated OMVs, but without the over-produced BAM – is missing even in the revised Extended Data Figure 4A,B. This must not only be included, but the findings from this experimental condition must also be discussed in the context of the findings from OmpT + SurA/Skp + colistin permeated OMVs + BAM, by comparing the read-outs of both experiments directly.

We think that the request by this referee was clear to us and the requested experiments were actually included in Extended Data Figure 4B: The referee asks that

OmpT + SurA/Skp + colistin-permeated OMVs without BAM overexpression

is compared with

OmpT + SurA/Skp + colistin-permeated OMV with BAM overexpression

The first is reaction 15 (and 17), and the second is reaction 11, with SurA as the chaperone of choice. We have done particular efforts to address this point by using two different preparations of OMVs without BAM overexpression, “Empty-OMVs” (reaction 15; these are endogenous OMVs) and once “mCherry-OMVs” (reaction 17; stimulated by overexpression of mCherry into the periplasm). As seen in the Figure, the activity of each of reaction 15 and 17 is negligible (<5%) compared to reaction 11 and this is also described and discussed in the manuscript text.

2. I disagree with the authors’ conclusion regarding the basal activity of OmpT. Can the authors clarify why the relative amounts of BamC and BamD in Extended Data Figure 4C is not in proportion with the BAM-OMVs sample of panel 4A? It also appears that in 4C, BAM-OMV samples containing Skp-OmpT and SurA-OmpT do not display the BAM protein bands in the same position.

We do not see what the referee finds here problematic. The relative amounts of BamC and BamD in Extended Data Figure 4C are well in agreement with panel 4A. Please see below Review-only-Figure 2, where we have arranged the relevant lanes of these two panels side-by-side.

ED4A shows BAM-OMVs, boiled and unboiled, on SDS-PAGE, where a volume of 3 μ L was loaded from a stock solution with 3.5 μ M BAM. There are three strong bands for BamA_F and BamC and BamD, and several weaker bands for other proteins. For panel ED4C, we were asked to load entire reaction mixes. In the reaction mixes, the BAM-OMVs are diluted to a concentration of 16 nM BAM, of which 7 μ L were loaded, i.e. the intensities are expected to be reduced by a factor of 94. In full agreement with this expectation, one can see the three strong bands of BamA/C/D as faint bands (red boxes in the sample with Skp-OmpT; the bands are also visible in the sample with SurA-OmpT). The bands come at the correct positions and their relative intensity is also maintained, within the precision of the SDS-PAGE at the undertaken dilution. This is all in agreement with expectations and we really do not see what the referee finds here problematic.

Review-only-Figure 2. (A) Arrangement of parts of panels A and C from Extended Data Figure 4. The lanes from each gel relevant for this discussion were arranged here side by side, such that the MW ladders align as best possible. Three red boxes were added to ED4A at the positions of BamA, BamC and BamD, and these three boxes were then copied into the corresponding position on the gel in ED4C. There, they come to match with 3 faint bands.

The basal activity measured as RFU in several reactions in 4B (=ED4B) are comparable to the highest amount of darobactin used in Extended Data Figure 6A. This again indicates that there is sufficient amounts of basal OmpT activity irrespective of BAM.

We disagree with this conclusion.

First, the activities presented in ED4B measured in RFU units are not directly comparable with the activities presented in ED6A, due to different composition of the respective reactions. In ED4B, there is 500 μ M QF and 16 nM BAM (defined in line 495), compared to 100 μ M QF and 10 nM BAM in ED6A (line 541), i.e. there is already a factor 8 difference expected.

Second, the activities in RFU are lower in ED6A than in ED4B, so that we readily reach a factor of 10 or more.

Third, while we agree that there remains a weak basal activity in some of the control reactions and at high concentrations of darobactin, this is however not at all a problem for the conclusions of the manuscript.

Therefore, I reiterate that in addition to checking these data, the authors must provide irrefutable evidence of 0% OmpT folding + activity in OMVs without BAM, or run at least one experiment with a protein other than OmpT.

Without these vital checkpoints in place, the study fails to address the most essential objective, which is to generate a functional BAM complex. Hence, the submission cannot be recommended in its current form.

We strongly disagree with the referee in this point. As outlined above and in the manuscript, we clearly do have achieved this objective. The bulk of the observed activity (>95%) results from BAM overexpression (see our answer to point 1 above) and the activity is sensitive to a known BAM inhibitor (Figure 2F, Figure 4B). Therefore, we have produced functional BAM in OMVs and the assay is suitable to high throughput screen for inhibitors. The residual basal activity does not change any of these conclusions.

Reviewer #2 (Remarks to the Author):

In their revision of “High-throughput screening of BAM inhibitors in native membrane environment”, Rath et al. have addressed the main concerns raised by the reviewers. Overall, this manuscript established a novel approach to screening for inhibitors of an essential activity in Gram-negative bacteria, optimized and validated this assay, and applied it to a test library to demonstrate its potential application. Especially critical to the revision was the inclusion of several additional control and optimization experiments for the assay (especially the data in Fig. 3 and Extended Data Figs. 4 and 5) and the expanded descriptions of the use of Darobactin and the initial small molecule screen. The additions and modification have greatly improved the manuscript and it is appropriate for the target audience of Nature Communications.

We thank this referee for the appreciation of our work.

Reviewers' Comments:

Reviewer #1:

Remarks to the Author:

The authors and I are clearly at a disagreement. To illustrate why their data is incomplete, I point out the following as an example:

In their rebuttal, the authors highlight three bands on a gel image they extracted from ED4C. They indicate that all three bands correspond to BAM components. If this is indeed true, why is the band highlighted in the middle box also seen in OmpTu and Skp-OmpTu samples in ED4C?

As I clearly stated in my previous comments, it is important that the authors check their data thoroughly. Additionally, they should maintain consistent sample conditions across their experiments!

Point-by-point response

REVIEWER COMMENTS

Reviewer #1 (Remarks to the Author):

In their rebuttal, the authors highlight three bands on a gel image they extracted from ED4C. They indicate that all three bands correspond to BAM components. If this is indeed true, why is the band highlighted in the middle box also seen in OmpT_U and Skp-OmpT_U samples in ED4C?

The reviewer refers to an arrangement of gels that was part of our previous rebuttal letter. For convenience, we reproduce these gels here again:

The three proteins in red boxes are BamA, BamC and BamD, from top to bottom. They are seen with high intensity in ED4A and with very weak intensity in ED4C, due to the applied dilution.

The reviewer now asks, why the band highlighted in the middle box (BamC) is also seen in the samples OmpT_U and Skp-OmpT_U in ED4C. So let us have a look at the uncropped SDS-PAGE of ED4C:

Supplementary Figure 1. Coomassie Blue stained 4–20% SDS-PAGE with molecular weight standard in the first lane. Samples were loaded with or without boiling, as indicated below the gel. Annotations of all bands are given on the right.

The sample $OmpT_U$ contains a single band (band 12). This is unfolded $OmpT$ at a MW of 35 kDa. This is not the same protein as band 4 in $BAM-OMV+Skp-OmpT$ (which was highlighted with a red box in the previous rebuttal letter). Band 4 is $BamC$ with a molecular weight of 36 kDa. The proteins run at nearly the same height due to their similar molecular weight, but can be distinguished.

So there is clearly no $BamC$ in the sample of $OmpT_U$.

Reviewers' Comments:

Reviewer #1:

None